# HDAC1: An Essential and Conserved Member of the Diverse Zn^2+^-Dependent HDAC Family Driven by Divergent Selection Pressure

**DOI:** 10.3390/ijms242317072

**Published:** 2023-12-02

**Authors:** Jing-Fang Yang, Le-Rong Shi, Ke-Chen Wang, Li-Long Huang, Yun-Shuang Deng, Mo-Xian Chen, Fang-Hao Wan, Zhong-Shi Zhou

**Affiliations:** 1State Key Laboratory for Biology of Plant Diseases and Insect Pests, Institute of Plant Protection, Chinese Academy of Agricultural Sciences, Beijing 100193, China; yangjingfang@caas.cn (J.-F.Y.); 82101222346@caas.cn (L.-R.S.); wangkechen1999@163.com (K.-C.W.); 15386163915@163.com (L.-L.H.); dys011128@163.com (Y.-S.D.); 2National Nanfan Research Institute (Sanya), Chinese Academy of Agricultural Sciences, Sanya 572024, China; 3National Key Laboratory of Green Pesticide, Key Laboratory of Green Pesticide and Agricultural Bioengineering, Ministry of Education, Guizhou University, Guiyang 550025, China; cmx2009920734@gmail.com; 4Shenzhen Branch, Guangdong Laboratory for Lingnan Modern Agriculture, Genome Analysis Laboratory of the Ministry of Agriculture and Rural Affairs, Agricultural Genomics Institute at Shenzhen, Chinese Academy of Agricultural Sciences, Shenzhen 518120, China

**Keywords:** Zn^2+^-dependent histone deacetylases, HDAC1, phylogenetic analysis, evolutionary plasticity, divergent selection pressures

## Abstract

Zn^2+^-dependent histone deacetylases (HDACs) are enzymes that regulate gene expression by removing acetyl groups from histone proteins. These enzymes are essential in all living systems, playing key roles in cancer treatment and as potential pesticide targets. Previous phylogenetic analyses of HDAC in certain species have been published. However, their classification and evolutionary origins across biological kingdoms remain unclear, which limits our understanding of them. In this study, we collected the HDAC sequences from 1451 organisms and performed analyses. The HDACs are found to diverge into three classes and seven subclasses under divergent selection pressure. Most subclasses show species specificity, indicating that HDACs have evolved with high plasticity and diversification to adapt to different environmental conditions in different species. In contrast, HDAC1 and HDAC3, belonging to the oldest class, are conserved and crucial in major kingdoms of life, especially HDAC1. These findings lay the groundwork for the future application of HDACs.

## 1. Introduction

Histone deacetylases (named HDACs in mammals) are enzymes that catalyze the removal of acetyl groups from lysine residues on histone proteins [1]. They are key epigenetic regulators that scaffold DNA packing into nucleosomes. HDACs are classified into two families based on the domain architecture and its dependence on co-factors: Zn^2+^- and nicotinamide adenine dinucleotide (NAD)-dependent HDACs. NAD-dependent HDACs are better known as the sirtuins protein family [2]. Zn^2+^-dependent HDACs (simplified as HDAC in this study), a classical and significant part of the HDAC family, have crucial biological functions. Their regulation is associated with numerous diseases, including cancer, developmental disorders, and neurodegenerative diseases [3]. In insects, they control the development by modulating hormone action [4]. For plants, they regulate development, cell cycle, programmed cell death, and stress response [5]. In fungi, they play a more prominent role in regulating the cell cycle progression, carbon metabolite, and carbohydrate transport and utilization [6]. Therefore, HDACs are essential in all biological kingdoms and have been identified as novel agents to treat cancers and potential targets for pesticides [7,8].

Until now, five drugs have been approved targeting HDACs: vorinostat, belinostat, romidepsin, tucidinostat, and panobinostat. They are the most clinically successful indications in the treatment of hematological neoplasms. Hydroxamic acid class agents all are pan-HDAC inhibitors. The first clinically successful HDAC inhibitor is vorinostat, which is an orally active inhibitor with hydroxamic acid used to treat cutaneous T-cell lymphoma (CTCL). Another two approved agents with the same pharmacophore are belinstat and panobinostat, which are used for the treatment of peripheral T-cell lymphoma (PTCL) and multiple myeloma (MM), respectively. The non-hydroxamic benzamides class includes tucidinostat and romidepsin, which are active Class 1 and 2 specific agents for use in PTCL and CTCL/PTCL [9]. Their scaffolds provide insight for the further HDAC inhibitors design.

HDACs are diverse across all kingdoms of life. They comprise 11 members in *Homo sapiens* (*H. sapiens*), but only have five members in *Saccharomyces cerevisiae* (*S. cerevisiae*) [10,11]. It indicates their complex evolutionary histories and different functions across kingdoms. As we all know, HDACs of *H. sapiens* are divided into four classes: Class I, Class IIa, Class IIb, and Class IV [10]. However, the classification of HDACs at the level of the whole biosphere is not clear. Meanwhile, many HDAC phylogenetic analyses have already been published, while the number of the used species is limited and these analyses only focused on individual genomes [5,6,12,13,14,15,16]. The holistic evolution of HDACs among the whole biological kingdom has not been reported.

In this study, we collected HDAC sequences from 1451 species and conducted repertoires, classifications, divergencies, and phylogenetic studies. We found that HDAC members vary among different species, and they are divided into three classes and seven subclasses: Class I (HDAC1, HDAC3, and HDAC8), Class II (HDAC4, HDAC6, and HDAC12), and Class IV (HDAC11). Each HDAC class shows its distinct features and functions. Class I and Class II contribute to the whole biological kingdoms, except for parasitic and symbiotic organisms and some lower species. While Class IV mostly exists in animals, green plants, and some lower organisms. Most subclasses have species specificity, indicating that HDACs have evolved with high plasticity and diversification. However, HDAC1 and HDAC3 are exceptions as they exist in all the major kingdoms of life. They belong to the oldest class of HDAC (Class I) but are the most conserved and essential members of the family, especially HDAC1 [17,18,19,20,21,22]. Moreover, the evolutionary mechanism of the HDACs divergence was revealed. We found that divergent selection pressure (DSP) is the main reason for the divergence of HDACs. The duplication event of HDAC is one of the reasons for their diversity in higher organisms. Together, they led to the diversification of HDACs. This work provides new perspectives on understanding the origin and roles of the HDAC family and lays a foundation for their application.

## 2. Results

### 2.1. The Distribution, Classification, and Phylogenetic Analysis of HDACs

In total, 11012 Zn^2+^-dependent *HDAC* genes encoding 18460 proteins were identified from 1451 species, including 392 animals (227 chordates, 162 insects, and three nematodes), 200 plants (151 flowering plants, 36 green algae, two charophyte algae, one uniseriate filamentous algae, one streptophyte algae, one fern, 6 mosses, and two other green plants), 733 fungi (374 ascomycete fungi, 247 basidiomycete fungi, two blastocladiomycetes, 26 chytrids, three glomeromycetes, 19 microsporidians, seven mucoromycota, 14 zoopagomycota, and 41 other fungi), 24 oomycetes, seven red algae, three proteobacteria, and 92 other species (Appendix A). We found that the number of *HDAC* genes (1–42 genes/1–75 proteins) varies widely among different species (Appendix A). *Thinopyrum intermedium* has the most HDAC, 42 family members, due to it being hexaploid wheat. The lower organisms have the fewest HDAC, such as *Entamoeba histolytica* (*E. histolytica*), *Spironucleus salmonicida*, *Cyberlindnera jadinii*, *Cryptomonas paramecium*, *Picocystis* sp. *ML*, and *Xanthomonas oryzae*. Among the major kingdoms, the median of *HDAC* gene numbers in animals is fewer than in green plants, but more than in fungi and oomycetes (Appendix A). Within animals, chordates have more *HDAC* genes than insects (Appendix A). For plants, the median of *HDAC* gene numbers of flowering plants is approximately equal to that of green algae. However, flowering plants have more points at large outliers in their gene number boxplot due to their multiploidy. Fungi have fewer HDACs than oomycetes, and basidiomycete fungi owe more HDACs than those of ascomycete fungi. Meanwhile, an extensive phylogenetic tree was developed by the Maximum Likelihood (ML) method to classify these *HDAC* genes into several subfamilies. Each clade contains *HDAC* genes from different kingdoms (Appendix A), indicating the complex origin of HDACs.

To further study the origin and diversification of *HDAC* genes, *HDAC* genes from typical species were collected for phylogenetic analysis. In total, 142 species were used in this section, including 38 animals (16 chordates, 20 insects, and two nematodes), 38 plants (34 flowering plants, one green alga, one fern, and two mosses), 36 fungi (20 ascomycete fungi, 11 basidiomycete fungi, two chytrids, one glomeromycetes, one microsporidians, and one mucoromycota), four oomycetes, one red alga, three proteobacteria, and 22 other species (Appendix A). One ML tree was constructed based on the HDAC protein sequences from these species. The clades are labeled based on the class names of the sequences from *H. sapiens* in the corresponding clade (Figure 1A and Appendix A). Class IIa and Class IIb are clustered in one big clade; thus, we grouped them together as Class II. Meanwhile, the sequences of each large group (Class I, Class II, and Class IV) were collected to build up evolution trees separately (Appendix A). These trees are similar to the phylogenetic trees published in the previous work [5,6,12,23]. In addition, the *HDAC* genes of eight selected species (*H. sapiens*, *Apis mellifera* (*A. mellifera*), *Arabidopsis thaliana* (*A. thaliana*), *Oryza sativa* (*O. sativa*), *Rhizoctonia solani* (*R. solani*), *S. cerevisiae*, *Pythium arrhenomanes ATCC 12531* (*P. arrhenomanes ATCC 12531*), *Phytophthora sojae* (*P. sojae*)) were used to construct an ML tree with higher precision to further reveal the rationality of the above tree (Appendix A). The consistency among these trees indicates that the ML tree built up using 142 selected species could be used for analysis. Moreover, the tree of eight species was also used in the following study.

Consequently, the HDACs are classified into three major classes (Class I, Class II, and Class IV) and seven subclasses (HDAC1, HDAC3, HDAC8, HDAC4, HDAC6, HDAC12, and HDAC11) according to the above trees (Figure 1B and Appendix A). The numbers of HDAC genes in green plants are all much more variable than in other major kingdoms (Appendix A). For HDACs in Class I, the median number of genes in the green plant (4) > animals (3) > oomycetes (2.5) > fungi (2), flowering plants (4.5) > green algae (3), chordates (4) > insects (3), basidiomycete fungi (4) > ascomycete fungi (2). For Class II, the medians of gene numbers of all classes of plants, fungi, and oomycetes are fixed at 5, 2, and 5.5. The corresponding number of animals, chordates, and insects are 2, 6, and 2, respectively. Class I and Class II contribute to the most species of major kingdoms, but Class I is not found in parasitic and symbiotic organisms, such as *Acidovorax citrulli*, *Xanthomonas oryzae*, *Pseudomonas syringae*, and *Symbiodinium microadriaticum*, and Class II is not found in some lower species, such as *Pseudomonas syringae*, *E. histolytica*, *Encephalitozoon hellem ATCC 50504*, and *Tritrichomonas foetus*. Class IV only exists in animals, green plants, and some lower organisms. The medians of *Class IV* genes number in the major groups of species are all equal to one, except for green algae (*Dunaliella salina*, 4). For subclasses of Class I, the medians of gene numbers of HDAC1 are much more than HDAC3 and HDAC8. The median of *HDAC1* gene numbers in green plants (3) > animals (2) > oomycetes (1) = fungi (1), flowering plants (3) > green algae (2), chordates (2) > insects (1), basidiomycete fungi (2) > ascomycete fungi (1). The medians of *HDAC3* gene numbers of all major groups of species are fixed at 1. HDAC8 is not found in plants, and its members are not too many in other species. For subclasses of Class II, HDAC4 does not exist in most fungi and oomycetes, but is enriched in animals and plants. HDAC6 is not found in oomycetes, while is widely contributing to other major groups of species. HDAC12 is not distributed in most animals, but plays a role in other species, especially oomycetes. HDAC11 is the only member of Class IV. These data indicate that HDACs play more varied roles in animals and plants than those in fungi, especially in higher organisms. Class I is much more diverse than Class II and Class IV within each major group of species, but it is not essential in parasitic and symbiotic organisms. For oomycetes, HDACs in Class II are much more important than other types of HDACs. HDACs in Class IV are not important in many biological kingdoms. For the subclasses, HDAC1 and HDAC3 play essential roles in all living organisms. The species specificities of other subclass members imply a high degree of evolutionary plasticity and functional diversification of HDACs in different species.

On the other hand, the phylogenetic relationship of HDACs can be revealed by the evolutionary trees (Figure 1A, Appendix A). Class I seems to be the oldest type of HDAC. HDAC8 is the root branch of HDAC1 and HDAC3, while it is lost in the green plant. HDAC1 is the newest member in Class I. Class II and Class IV are derived from the same origin with Class IV being fresher than Class II. HDAC12 is the root of HDAC4 and HDAC6.

### 2.2. The Divergence of HDACs

The HDAC function can be characterized by the sequences and structures of genes and proteins. *HDAC* gene structures of each class do not have their typical characteristics (Appendix A). However, the protein and cds motifs of Class I are significantly different from Class II and Class IV. While there is no significant difference among subclasses in each major class (Figure 2 and Appendix A), the shorter sequences of Class IV lead to its lesser motifs. Meanwhile, the Pfam domains were also analyzed to feature the HDAC protein sequences (Figure 2). The Histone deacetylase (Hist_deacetyl) domain is the only conserved domain among these proteins. We extracted the corresponding sequences of the final selected eight species to carry out analysis. The phylogenetic tree built using Hist_deacetyl domain sequences is very similar to the one based on the whole sequences, but with some small differences (Appendix A). The clades of the three major classes (Class I, Class II, and Class IV) are not changed. This indicates the evolutionary consistency of the whole sequences and Hist_deacetyl domain sequences. Further, the protein sequence identities of these three classes were calculated (Appendix A). The identities of Hist_deacetyl domain sequences are higher than those of the whole sequences. The identity of Class I sequences is higher than other classes, implying that Class I is much more conserved than others. Therefore, Class I, Class II, and Class IV have their features, but the characteristics of subclasses are not significant at the sequence level. Even though Class I is the older member, it is most conserved among all the classes.

To visualize the evolutionary conservation profiles of HDACs in different classes, their ConSurf grades were calculated based on the sequences from 1451 species. The results are shown through human HDAC2 (belongs to Class I and HDAC1) complexed with vorinostat (colored in yellow) crystal structure, colored from blue (low grade) to red (high grade) (Appendix A). Vorinostat coordinates the Zn atom and forms hydrogen bonds with Asp100, His141, and Tyr304. The residues within the 6 Å distance to its binding ligand with low ConSurf grade (≤7) are shown in sticks. The residues for Zn atom binding (Asp177, His179, and Asp265) and for catalyzing histone deacetylase (His141, His142, and Tyr304) are highly conserved. Hence, their functions for histone deacetylase are conserved [5]. Gly28 and Glu99 are in low ConSurf grades in all the HDAC classes. For three major classes, some residues of Class I are lost in Class IV, even the residues near vorinostat, such as Phe206 and Glu99. Class I (4) has fewer residues in low ConSurf grades than Class II (7). For subclasses of Class I, HDAC1 (5) has more residues in low ConSurf grade than HDAC3 (3), but fewer than HDAC8 (8). For subclasses of Class II, some gaps (colored in yellow) are found in each subclass, and the gap residues of HDAC4 and HDAC12 are near the binding ligand. In summary, the binding domain of Class I is most complete and conserved, especially HDAC1 and HDAC3. It may be the reason for their higher drug ability in humans [9]. Meanwhile, the difference in subclasses of HDACs may be used for highly selective drug or pesticide design.

Transcriptomic data can serve as an indicator of physiological and tissue-specific function. Therefore, the gene expression data of HDACs in *H. sapiens*, *A.mellifera*, *A. thaliana*, *O. sativa*, and *S. cerevisiae* in different developmental stages and under different treatments were tested, collected, and analyzed (Figure 3 and Appendix A). Interestingly, at least one member of HDAC1 or HDAC3 is highly expressed in these species, especially HDAC1. They are up-regulated in the cancerous tissue of *H. sapiens*, heat treated *S. cerevisiae*, cold or drought treated seedlings of *A. thaliana* (Figure 3), and ABA treated shoot of *O. sativa* (Appendix A). While they are down-regulated in *S. cerevisiae* in the lower pH environment and ABA treated seedlings of *A. thaliana* (Figure 3 and Appendix A). Further, HDAC1 plays an essential role in many diseases. It provides opportunities for cancer treatment [24], the aging brain and Alzheimer’s disease repairment [21], mental disease cure [25], sickle cell disease treatment [26], long-lasting social fear extinction [27], and so on. Therefore, HDAC1 is essential for living organisms and human health.

As we know, HDACs always work through forming complexes with repressor proteins [28]. Therefore, the proteins interacting with the important HDAC1 members, including ENSP00000362649.3 (HDAC1) of *H. sapiens*, YNL330C (RPD3) of *S. cerevisiae*, and AT4G38130.1 (HDA19) of *A. thaliana*, are collected from the STRING database. Meanwhile, their formed complexes are shown in Figure 4. It appears that the functions of the interacted proteins in *H. sapiens* and *S. cerevisiae* are studied more thoroughly than in *A. thaliana*. There are many complexes performing the function of histone deacetylase, while most of them have species specificity. For example, NuRD complex and SWI/SNF superfamily-type complex are found in *H. sapiens*, while Rpd3 related complexes and Snt2C complex in *S. cerevisiae*. However, the Sin3-type complex is found and studied in all these species. Sin3 complexes play important roles in facilitating local histone deacetylation to regulate chromatin accessibility and gene expression [29]. From above, we know that HDAC1 in many species may execute their functions in the same way.

### 2.3. The Gene Duplication, Molecular Evolutionary, and Functional Divergence Analysis of HDACs

To understand the mechanism for gene divergence, the synteny and collinearity of HDAC in selected eight species were first analyzed (Figure 5A). The collinear duplicated event is only found in *H. sapiens*: ENSP00000362649.3 and ENSP00000430432.1 in HDAC1; ENSP00000225983.5 and ENSP00000408617.2 in HDAC4. Tandem duplicated genes are found in *H. sapiens* (ENSP00000362669.3 and ENSP00000497072.1 in HDAC8), *A. thaliana* (AT5G61060.2 and AT5G61070.1 in HDAC6), *O. sativa* (LOC_Os05g36920.1 and LOC_Os05g36930.1 in HDAC12), and *P. arrhenomanes ATCC 12531* (EPrPR00000013832 and EPrPR00000013833 in HDAC12). The results reveal that the HDAC duplication event is not common in all the species, but only in the advanced organisms. This might be the reason for the high diversity of HDACs in the higher organisms.

Selection pressure is always the force that drives gene divergence. Class I, Class II, and Class IV have their distinctive features. Hence, we estimated the ratio of non-synonymous to synonymous substitutions (ω = dN/dS) of Class I, Class II, and Class IV under different codon substitution-based evolutionary models. The high correlation between the sequence (Class I and Class II) conservations of the selected eight species and the 142 representative species implies the HDAC sequence of eight selected species could be used to represent the most species (Appendix A). From the above, we know that Class II and Class IV originate from the same root, and Class I and Class II are two main classes. Hence, the phylogenetic trees of Class I, Class II, and Class IV, and Class I and Class II in the eight selected species shown in Appendix A were used for this analysis. The mean nonsynonymous (dN) and synonymous (dS) substitution rates ω (dN/dS) value of Class I, Class II, and Class IV is 0.11001 (Appendix A). The two-ratio branch model indicates a faster evolution for the foreground branches of Class II and Class IV (ω = 0.116512) than Class I (ω = 0.0904145). The branch-site model indicates strong DSP of classes with heterogeneous ω values across sequences and branches (Model 3 vs Clade model D), which leads to each class paralog subfamily. However, no residue under positive selection (PS) is predicted significantly using the branch-site models for the PS analysis (Model A null vs Model A). The results for Class II vs Class I are similar to Class II and Class IV vs Class I (Table 1). The mean ω value of Class I and Class II is 0.124240, and the evolutionary speed of Class II (ω = 0.154068) is faster than Class I (ω = 0.112429). The DSP is also found, and no positive selected site is detected in these two classes. Therefore, DSP is the main reason for the divergence of HDACs.

To understand whether the major classes (Class I and Class II) derived from DSP experience functional divergence (FD) during evolution, the type I FD between Class I and Class II was analyzed. The results show that the coefficient of FD I (θ) is 0.3888 ± 0.063 between Class I and Class II, suggesting their significantly different functions (Table 2). Twelve critical amino acid sites responsible for FD I are identified. Some amino acid residues in these sites are remarkably conserved in one of the classes, but are variable and different in physicochemical features in another class (Figure 5B, Appendix A). The residues at site 137 of human HDAC2 are not conserved in Class I, but are conserved (Pro) in Class II. The residues at the sites 153, 187, and 232 of human HDAC2 are conserved in Class I (Tyr, Ala, and Gly), but are not conserved in Class II. These residues must induce some difference between these two classes. The type I FD residues, Ala187 and Arg230 in human HDAC2 (PDBID: 8c60) form hydrogen bonds with paired amphipathic helix protein Sin3b (SIN3B) and PHD finger protein 12 (PHF12), respectively (Appendix A). Hence, the changes in the residues at these two sites and their near residue (Phe188) must affect Sin3 complex formation to influence the function of Class I and Class II. Meanwhile, all the human HDAC members in Class II indeed could not form the Sin3 complex [30]. The residues at site 178 of HDAC2 in human is the only residue within the 6 Å distance to the binding ligand (Figure 5B). The lengths of their side chains might influence their function directly. The corresponding residues of the 178 in human HDAC2 of some HDAC members in *R. solani* are absolutely different from other species, suggesting their difference for catalysis. The results of FD analyses reveal the underlying mechanism of different functions of HDACs in different classes and species.

## 3. Discussion

Zn^2+^-dependent HDACs are widely distributed in biology and play central roles in living organisms. However, their repertoires, classifications, phylogenetic relationships, divergencies, and evolutionary mechanisms across different kingdoms are not fully understood. The present study was conducted to address these questions using the corresponding HDAC sequences from 1451 species. HDAC members vary widely among different species and are divided into three classes and seven subclasses: Class I (includes HDAC1, HDAC3, and HDAC8), Class II (includes HDAC4, HDAC6, and HDAC12), and Class IV (HDAC11). Class I, Class II, and Class IV each have their own unique features. Class I and Class II are found throughout the biological kingdom, but Class I is not found in parasitic and symbiotic organisms, and Class II is not found in some lower species. Class IV mostly exists in animals, green plants, and some lower organisms. Most subclass members have species specificities. However, HDAC1 and HDAC3 play essential roles in all living organisms. They belong to the oldest class of HDACs (Class I), which are the most conserved among all the classes. Meanwhile, their binding domains are the most complete and have the highest evolutionary conservation. Further, the transcriptome data revealed that HDAC1 and HDAC3 play the most essential biological function among all the species, especially HDAC1. In addition, we found that HDAC1 in many species could play their functions through the Sin3 complex, indicating that they could work in a conservative way in various species. On the other hand, the evolutionary mechanism of the HDACs divergence was revealed. We found that DSP is the main reason for the divergence of HDACs. The duplication event of HDAC is one of the reasons for their diversity in higher organisms. Moreover, the results of FD analyses reveal that the residues at the sites 137, 153, 178, 187, 188, 230, and 232 of HDAC2 in humans may determine different functions of HDACs in Class I and Class II and in different species. In a word, under DSP, living organisms in different kingdoms diverge into many unique class members to adapt to their environments, while HDAC1 is still conserved and plays its central role in all species. These results provide new insights for understanding the origin and roles of HDAC family members and will guide future studies in this field.

## 4. Materials and Methods

### 4.1. HDAC Gene Related Data Collection and Analysis

The BLASTp tool was used for the searching and collection of protein sequences similar to the ENSP00000362649.3 (HDAC1) of *H. sapiens* in the species from Phytozome v13.0, Ensembl, Ensemble Plant, Ensembl Fungi, Ensembl Protists, and the National Center for Biotechnology Information (NCBI) databases [31,32,33,34,35]. The coding sequence, gene structures, and other related data were retrieved from these databases. The domains of sequences were predicted using profile hidden Markov Models from the Pfam database through HMMER v3.1b2 software [36]. Then, only the sequences with Hist_deacetyl (PF00850) domain were maintained. The motifs (MEME) analysis was carried out using TBtools v1.120 [37]. The gene synteny, collinearity, and duplication were detected by the MCScanX v1.0 [38].

### 4.2. Phylogenetic Analysis

The longest-retained coding proteins for genes were used to perform multiple protein sequence alignment using MEGA6 v6.0 software [39]. The trees were both built up based on the HDAC protein sequences from 1451 species and the 142 selected species by FastTree v2.1.9 [40]. All the ML trees with higher precision were constructed using PhyML v3.0 [41]. Their best models were all determined by using the Akaike Information Criterion (AIC) and the Bayesian Information Criterion (BIC) based on ProtTest 3 [42,43]. The final ML trees were represented and edited using FigTree v1.4.3 [44].

### 4.3. Evolutionary Conservation Analysis

The human HDAC2 crystal structures complex with vorinostat (PDBID: 4lxz) were obtained from RCSB Protein Databank (PDB) and were used to present the Consurf Grade calculated through the ConSurf Web server [45,46]. The models were prepared using Pymol v2.4 software [47].

### 4.4. RNA-Seq Data Collection and Analysis

The gene expressions of HDACs of *H. sapiens*, *A. mellifera*, and *S. cerevisiae* were analyzed and collected using the Geo2R tool merged in the NCBI database. The data in normal salivary gland tissue (NSG)/adenoid cystic carcinoma (ACC) of *H. sapiens* (GEO accession: GSE153230), in bladder carcinoma tissue (BCT)/normal tissue (NT) of *H. sapiens* (GEO accession: GSE236932), in *A. mellifera* installed into eggplant greenhouses after 0 days, 14 days, 42 days, and 70 days (GEO accession: GSE29252), in *S. cerevisiae* with (39 °C) or without heat stress (30 °C) (GEO accession: GSE33276), and in *S. cerevisiae* treated with pH 2.5 and pH 4.5 were used. The RNA-Seq data of *HDAC* genes in *A. thaliana* (Project ID: PRJNA494179) and *O. sativa* (Project ID: PRJDA46487) under ABA, cold, and drought treatment in different tissues were downloaded from Arabidopsis RNA-seq Database [48] and Rice RNA-seq Database [49], respectively. The data were organized and shown using R v4.1.2 software [50].

### 4.5. Analysis of HDAC1 Physical Interaction

The search tool STRING (https://string-db.org/) (accessed on 1 December 2023) was used to create the protein physical interaction networks of ENSP00000362649.3 of *H. sapiens*, YNL330C of *S. cerevisiae*, and AT4G38130.1 of *A. thaliana*. We chose no more than fifty interacted genes determined by the experiments. The interactions with confidence of more than 0.4 are shown. The information about the complex is annotated by the Go databases.

### 4.6. Selection Pressure Analysis

The CodeML program of PAML v4.9 was used to estimate ω and to detect PS in HDAC coding genes [51]. Branch models and branch-site models were used to fit each orthologous group. Branch models are always used for detecting PS on a particular branch, allowing the ω ratio to vary among them. Branch-site models not only allow the defined foreground branches and background branches to have different ω values, but also can be used to predict the selective position among the sequences. The ω ratios and a log-likelihood value estimated by the ML method were used to examine the models (one-ratio model 0 vs two-ratio model 2; model A null vs model A; model 3 vs clade model D) based on the aligned coding sequences and tree topologies. The branch one-ratio model 0 maintains all branches in the same ω ratio, while a two-ratio model 2 allows ω to vary between the foreground and the background branches. The branch-site model A allows a class of sites in foreground branches under PS (ω > 1), but all the ω values are fixed in the model A null model. Bayes Empirical Bayes (BEB) method was used to calculate posterior probabilities of a codon under PS in Model A. Model 3 allows ω to vary among sites, but keeps ω constant among branches. The branch-site model D allows a class of sites to be under DSP between the foreground and the background branches.

### 4.7. Functional Diversification Analysis

DIVERGE v3.0 software was used to analyze the FD among the branches based on the selected protein sequences [52]. The type I functional diversification is used to reveal the occurrence of altered functional constraints between protein groups.

## Figures and Tables

**Figure 1 ijms-24-17072-f001:**
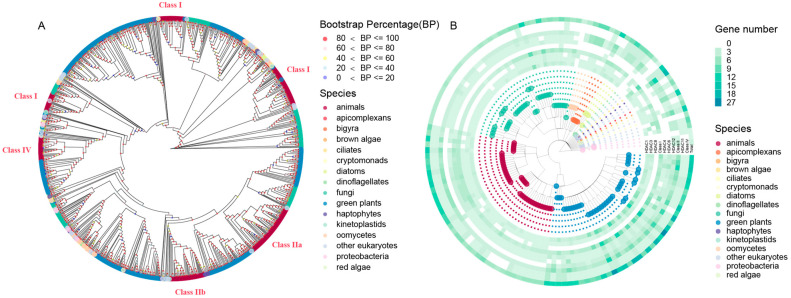
The phylogenetic tree and repertoires of HDACs from the 142 selected species. The Maximum Likelihood (ML) phylogenetic tree was built up using FastTree (**A**). The circle nodes are colored by the bootstrap percentage, the outer ring is colored by the species. Its enlarged tree is shown in Appendix A. The clades were labeled based on the class names of the sequences from *H. sapiens* in the corresponding clade. The statistical data for each class in these species is shown in (**B**). The circle nodes and dots are colored by the species. The corresponding detailed data is shown in Appendix A. The common tree of the selected species was obtained from the NCBI database. These figures were prepared using the ggtree package in R programming.

**Figure 2 ijms-24-17072-f002:**
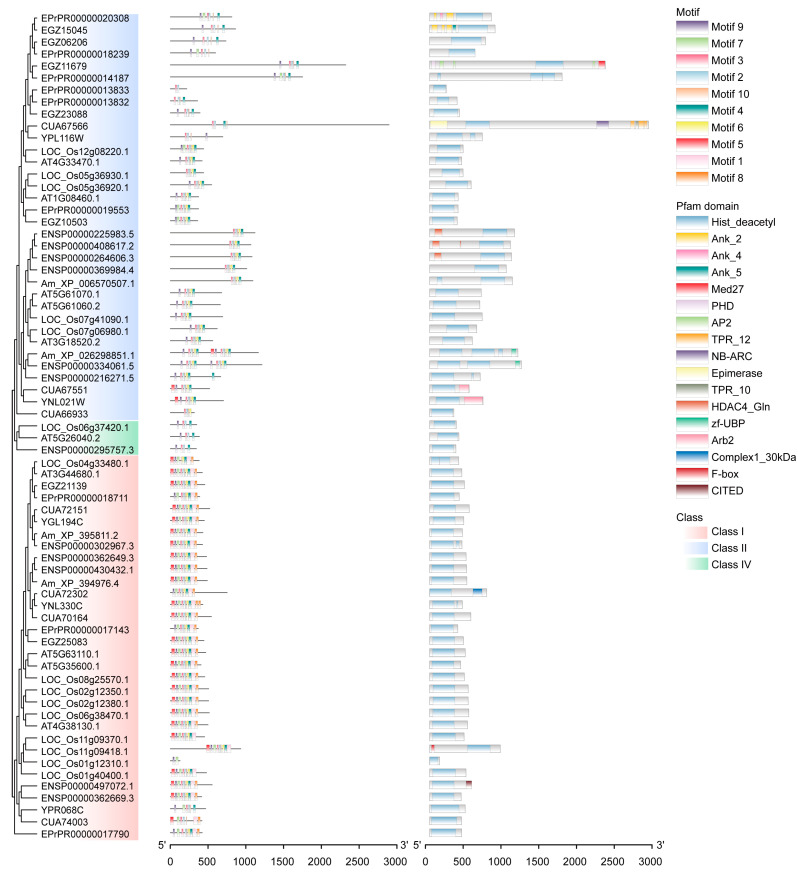
The motifs and domains of HDAC protein sequences. The motifs and domains were annotated using the TBtools v1.120 and the HMMER v3.1b2 software, respectively. The protein motifs are shown in Appendix A.

**Figure 3 ijms-24-17072-f003:**
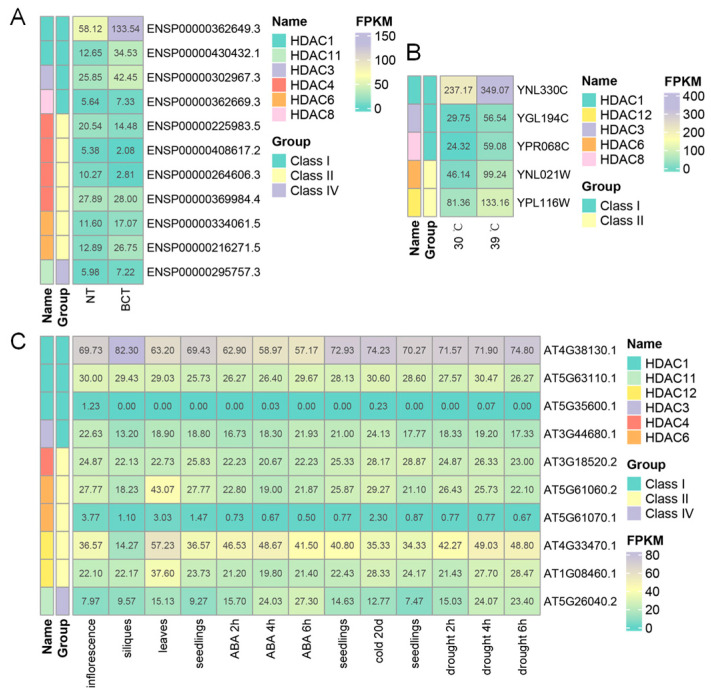
The gene expressions of HDACs in *H. sapiens*, *S. cerevisiae*, and *A. thaliana*. The data in bladder carcinoma tissue (BCT)/normal tissue (NT) of *H. sapiens* (**A**), in *S. cerevisiae* with (39 °C) or without heat stress (30 °C) (**B**), and in *A. thaliana* seedlings under ABA, cold, and drought treatment, and in other tissues (**C**) are shown by heatmap.

**Figure 4 ijms-24-17072-f004:**
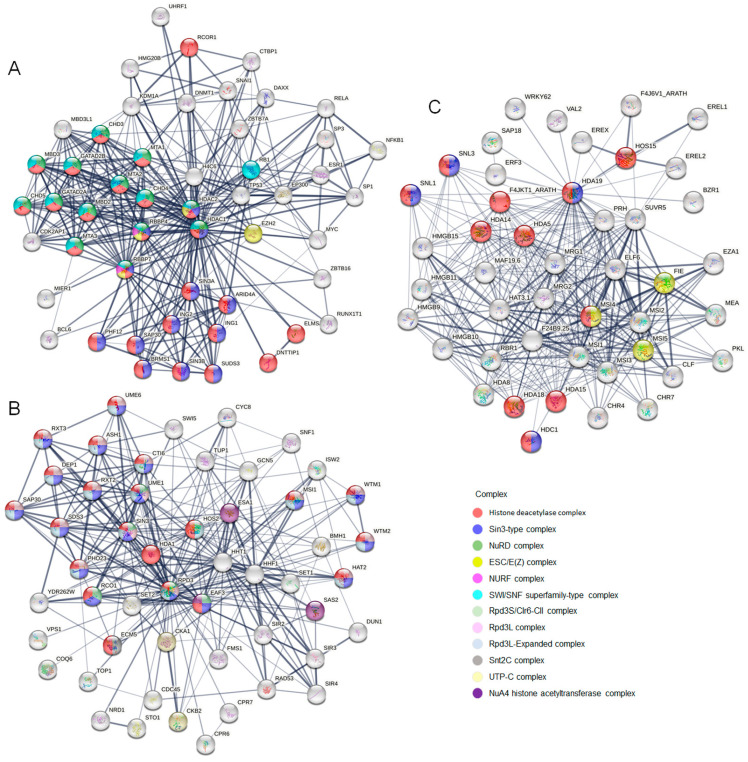
The genes formed physical complexes with HDAC1. Proteins interacting with ENSP00000362649.3 (HDAC1) of *H. sapiens* (**A**), YNL330C (RPD3) of *S. cerevisiae* (**B**), and AT4G38130.1 (HD1) of *A. thaliana* (**C**) are shown as bubbles. Their different colors indicate the complexes they formed, which are listed on the right. The line thickness indicates the strength of data support.

**Figure 5 ijms-24-17072-f005:**
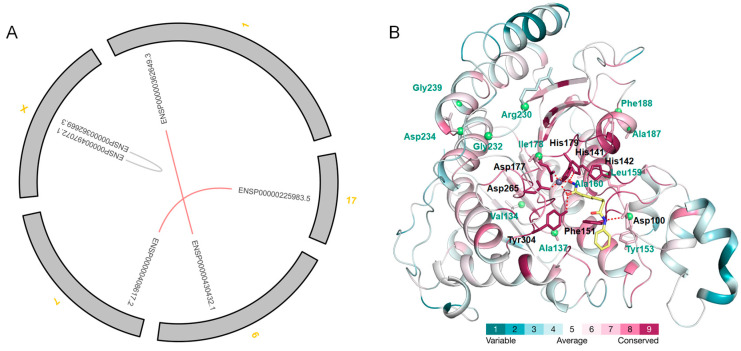
The synteny and collinearity of HDACs from *H. sapiens* and the 3D structure of *H. sapiens* HDAC2 showing functional divergence (FD). The collinear and tandem duplicated genes are calculated using MCScanX software and are labeled using red and gray lines, respectively (**A**). 1, 6, 7, 17, and X represent 1, 6, 7, 17, and X Chromosomes of *H. sapiens*. The CA atoms of residues involved in FD among Class I and Class II are shown as green balls (**B**). The human HDAC2 crystal structures complex with vorinostat (PDBID: 4lxz) are colored according to the degree of conservation (ConSurf grade) from blue (low grade) to red (high grade). The complexed ligand (vorinostat) and important residues are shown in sticks. The ConSurf grades of residues and the FD sites were calculated using Class I and Class II sequences.

**Table 1 ijms-24-17072-t001:** Parameter estimates, Ln L values, and LRTs of codon substitution evolutionary analyses of selective patterns for Class I and Class II members of the selected HDACs.

Foreground Branch	Model	*p* ^a^	Parameter Estimates					Ln L	*p*	BEB ^b^
			(Frequency, f, and ω Values)							
Class II	One-ratio model 0 (ω_0_ = ω_1_)	128	ω_0_ = ω_1_ = 0.1242					−138,343.69	3.08 × 10^−4^	NA
	Two-ratio model 2 (ω_0_, ω_1_)	129	ω_0_ = 0.1124; ω_1_ = 0.1541					−138,337.18		NA
	Model 3 (discrete)	132	-					−134,286.29	2.15 × 10^−4^	NA
	Clade model D (K = 3)	133	site class	0		1	2	−134,279.44		NA
			proportion	0.1031		0.5918	0.3050			
			branch type 0	0.0274		0.4668	0.0974			
			branch type 1	0.0274		0.4668	0.1185			
	Model A null (ω_2_ = 1)	130	1					−135,334.52	1	NA
	Model A (0 < ω_0_ < 1)	131	site class	0	1	2a	2b	−135,334.52		7 **, 9 **, 10 **, 28 **, 50 **, 53 *, 56 **, 67 **, 84 *, 88 **, 96 **, 251 **, 324 **, 327 *, 328 **, 329 **, 332 **, 368 **, 376 **, 377 **, 379 *, 380 **, 381 **, 382 **, 383 **, 384 **, 385 **, 386 **, 393 **, 396*, 397 **, 398 *
			proportion	0.2656	0.2667	0.2334	0.2343			
			background w	0.0944	1.0000	0.0944	1.0000			
			foreground w	0.0944	1.0000	1.0000	1.0000			

^a^ Number of parameters in the ω distribution. ^b^ Amino acids from BEB analysis as fixed by PS with posterior probabilities (>95% indicated with one asterisk, >99% indicated with two asterisks) are shown. The numbers corresponding to the positions in HDAC2 (ENSP00000430432.1) of *H. sapiens* were shown in Appendix A.

**Table 2 ijms-24-17072-t002:** Analysis of functional divergence (FD) for Class I and Class II members of the selected HDACs.

HDAC Subfamily	Coefficient θ_I_	θ_SE_	Critical Amino Acid Site (Q_k_ > 0.7)	*p*
Class I vs Class II	0.3888	0.063	134, 137 **, 153 **, 159, 160, 178, 187 **, 188, 230 **, 232 *,234,239 *	1.78 × 10^−6^

Critical amino acid sites related to FD with *p* > 70% (>90% indicated with one asterisk, >95% indicated with two asterisks). Amino acid from BEB analysis of PS was indicated with an underline. The numbers corresponding to the positions in HDAC2 (ENSP00000430432.1) of *H. sapiens* are shown in Appendix A.

## Data Availability

Data is contained within the article.

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
