# Peer review of "HDAC1: An Essential and Conserved Member of the Diverse Zn^2+^-Dependent HDAC Family Driven by Divergent Selection Pressure"

_ijms, 2023, doi:10.3390/ijms242317072_

Round 1
Reviewer 1 Report
Comments and Suggestions for Authors
The manuscript studies the evolution of HDACs (and specially HDAC1) between species. The methods are well described and results are interesting. However I have some concerns:
1.- Introduction: Authors are focused in HDAC1 because it is the HDAC more conserved and essential. They should add some previous bibliography to reforce this sentence
2.- A explanation of HDAC1 functions and its role in different disease would be informative
3.- Bibliography is short, authors should try to add more references about the topic.
Author Response
Dear Prof. Maurizio Battino,
Thanks for your efforts in the manuscript ijms-2729315, entitled ‘HDAC1: An Essential and Conserved Member of the Diverse Zn2+-Dependent HDAC Family Driven by Divergent Selection Pressure’, which we previously submitted for consideration of publication in International Journal of Molecular Sciences. We appreciate you and the reviewers for the helpful comments and suggestions, which can help us to improve our manuscript. The manuscript has been revised in response to your and the reviewers’ suggestions. Please notice below our responses to the reviewers’ comments. Here is a summary of our changes and responses point by point:
RE Comments from Reviewer #1:
Comments 1: The manuscript studies the evolution of HDACs (and specially HDAC1) between species. The methods are well described and results are interesting. However, I have some concerns.
Answer: Thanks for your positive comments on our manuscript. We have already revised our paper according to your suggestions.
Comments 2: Introduction: Authors are focused in HDAC1 because it is the HDAC more conserved and essential. They should add some previous bibliography to reinforce this sentence.
Answer: We have added bibliographies (18-23) to reinforce this sentence following:
- He, Y.; Petrie, M. V.; Zhang, H.; Peace, J. M.; Aparicio, O. M., Rpd3 regulates single-copy origins independently of the rDNA array by opposing Fkh1-mediated origin stimulation. Proc. Natl. Acad. Sci. U. S. A. 2022, 119, (40).
- Lee, S. H.; Farh, M. E.-A.; Lee, J.; Oh, Y. T.; Cho, E.; Park, J.; Son, H.; Jeon, J., A Histone Deacetylase, Magnaporthe oryzae RPD3, Regulates Reproduction and Pathogenic Development in the Rice Blast Fungus. Mbio 2021, 12, (6).
- Gregoricchio, S.; Polit, L.; Esposito, M.; Berthelet, J.; Delestre, L.; Evanno, E.; Diop, M. B.; Gallais, I.; Aleth, H.; Poplineau, M.; Zwart, W.; Rosenbauer, F.; Rodrigues-Lima, F.; Duprez, E.; Boeva, V.; Guillouf, C., HDAC1 and PRC2 mediate combinatorial control in SPI1/PU.1-dependent gene repression in murine erythroleukaemia. Nucleic Acids Res. 2022, 50, (14), 7938-7958.
- Yang, Y.; Yang, C.; Li, T.; Yu, S.; Gan, T.; Hu, J.; Cui, J.; Zheng, X., The Deubiquitinase USP38 Promotes NHEJ Repair through Regulation of HDAC1 Activity and Regulates Cancer Cell Response to Genotoxic Insults. Cancer Res. 2020, 80, (4), 719-731.
- Pao, P.-C.; Patnaik, D.; Watson, L. A.; Gao, F.; Pan, L.; Wang, J.; Adaikkan, C.; Penney, J.; Cam, H. P.; Huang, W.-C.; Pantano, L.; Lee, A.; Nott, A.; Phan, T. X.; Gjoneska, E.; Elmsaouri, S.; Haggarty, S. J.; Tsai, L.-H., HDAC1 modulates OGG1-initiated oxidative DNA damage repair in the aging brain and Alzheimer's disease. Nat Commun 2020, 11, (1).
- Zong, W.; Ren, D.; Huang, M.; Sun, K.; Feng, J.; Zhao, J.; Xiao, D.; Xie, W.; Liu, S.; Zhang, H.; Qiu, R.; Tang, W.; Yang, R.; Chen, H.; Xie, X.; Chen, L.; Liu, Y.-G.; Guo, J., Strong photoperiod sensitivity is controlled by cooperation and competition among Hd1, Ghd7 and DTH8 in rice heading. New Phytol. 2021, 229, (3), 1635-1649.
Comments 3: A explanation of HDAC1 functions and its role in different disease would be informative.
Answer: This is a good suggestion. We have added the corresponding description following “Further, HDAC1 plays an essential role in many diseases. It provides opportunities for cancer treatment [25], the aging brain and Alzheimer's disease repairment [22], mental disease cure [26], sickle cell disease treatment [27], long-lasting social fear extinction [28], and so on. Therefore, HDAC1 is essential for living organisms and human health.”. Furthermore, the content about HDAC inhibitors has already been added following “Until now, five drugs have been approved targeting HDACs: vorinostat, belinostat, romidepsin, tucidinostat, and panobinostat. They are the most clinically successful indications in the treatment of hematological neoplasms. Hydroxamic acid class agents all are pan-HDAC inhibitors. The first clinically successful HDAC inhibitor is vorinostat, which is an orally active inhibitor with hydroxamic acid used to treat cutaneous T-cell lymphoma (CTCL). Another two approved agents with the same pharmacophore are belinstat and panobinostat, which are used for the treatment of peripheral T-cell lymphoma (PTCL) and multiple myeloma (MM), respectively. The non-hydroxamic benzamides class includes tucidinostat and romidepsin, which are active Class 1 and 2-specific agents for use in PTCL and CTCL/PTCL [9]. Their scaffolds provide insight for the further HDAC inhibitors design.”
Comments 4: Bibliography is short, authors should try to add more references about the topic.
Answer: According to your suggestions, we finally cited 53 references about the topic.

Reviewer 2 Report
Comments and Suggestions for Authors
The manuscript describes the evolution and appearance of crucial HDAC enzymes in living organisms. The results are well described and presented. Some minor issues should be dealt with in a revised version of the manuscript:
Introduction: Please provide brief info about class III non-zinc HDACs (sirtuins), although they are not covered by the manuscript. Please also mention that some HDAC inhibitors such as vorinostat were already approved as drugs for human diseases in the introduction.
Table 1: Please modify the incorrect word separation in ´´Foregroun – d´´ (left column).
Line 334: Remove ´´all´´.
Line 335: The meaning of ´´living lives´´ is unclear to me.
Line 348: Replace ´´… environments. While HDAC1 …´´ by ´´… environments, while HDAC1 …´´.
Discussion: Please discuss the described interaction of vorinostat with HDACs. The relevance of the provided data for the development of drugs and pesticides should be outlined.
Comments on the Quality of English Languagen.a.
Author Response
Dear Prof. Maurizio Battino,
Thanks for your efforts in the manuscript ijms-2729315, entitled ‘HDAC1: An Essential and Conserved Member of the Diverse Zn2+-Dependent HDAC Family Driven by Divergent Selection Pressure’, which we previously submitted for consideration of publication in International Journal of Molecular Sciences. We appreciate you and the reviewers for the helpful comments and suggestions, which can help us to improve our manuscript. The manuscript has been revised in response to your and the reviewers’ suggestions. Please notice below our responses to the reviewers’ comments. Here is a summary of our changes and responses point by point:
RE Comments from Reviewer #2:
Comments 1: The manuscript describes the evolution and appearance of crucial HDAC enzymes in living organisms. The results are well described and presented. Some minor issues should be dealt with in a revised version of the manuscript.
Answer: Thanks for your positive comments on our manuscript, and we have already revised our manuscript according to your suggestions.
Comments 2: Introduction: Please provide brief info about class III non-zinc HDACs (sirtuins), although they are not covered by the manuscript. Please also mention that some HDAC inhibitors such as vorinostat were already approved as drugs for human diseases in the introduction.
Answer: We have added a brief info about class III non-zinc HDACs (sirtuins) following “HDACs are classified into two families based on the domain architecture and its dependence on co-factors: Zn2+ and nicotinamide adenine dinucleotide (NAD) -dependent HDACs. NAD-dependent HDACs are better known as the sirtuins protein family [2].”. We also mentioned the HDAC inhibitors approved as drugs for human diseases in the introduction following “Until now, five drugs have been approved targeting HDACs: vorinostat, belinostat, romidepsin, tucidinostat, and panobinostat. They are the most clinically successful indications in the treatment of hematological neoplasms. Hydroxamic acid class agents all are pan-HDAC inhibitors. The first clinically successful HDAC inhibitor was vorinostat, which is an orally active inhibitor with hydroxamic acid used to treat cutaneous T-cell lymphoma (CTCL). Another two approved agents with the same pharmacophore are belinstat and panobinostat, which are used for the treatment of peripheral T-cell lymphoma (PTCL) and multiple myeloma (MM), respectively. The non-hydroxamic benzamides class includes tucidinostat and romidepsin, which are active Class 1 and 2-specific agents for use in PTCL and CTCL/PTCL [9]. Their scaffolds provide insight for the further HDAC inhibitors design.”.
Comments 3: Table 1: Please modify the incorrect word separation in ´´Foregroun – d´´ (left column).
Answer: We have modified “Table 1” according to your suggestion.
Comments 4: Line 334: Remove ´´all´´.
Answer: It has already been removed.
Comments 5: Line 335: The meaning of ´´living lives´´ is unclear to me.
Answer: We are sorry to make it unclear. “living lives” means “living organisms”, and we have already replaced it in the manuscript following “For the subclasses, HDAC1 and HDAC3 play essential roles in all living organisms; However, HDAC1 and HDAC3 play essential roles in all living organisms”.
Comments 6: Line 348: Replace ´´… environments. While HDAC1 …´´ by ´´… environments, while HDAC1 …´´.
Answer: Thanks for your suggestions. We have already corrected it.
Comments 7: Discussion: Please discuss the described interaction of vorinostat with HDACs. The relevance of the provided data for the development of drugs and pesticides should be outlined.
Answer: This is a good suggestion. We have already added a responding discussion in the manuscript following “To visualize the evolutionary conservation profiles of HDACs in different classes, their ConSurf grades were calculated based on the sequences from 1451 species. The results are shown through human HDAC2 (belongs to Class I and HDAC1) complexed with vorinostat (colored in yellow) crystal structure, colored from blue (low grade) to red (high grade) (Figure S9). Vorinostat coordinates the Zn atom and forms hydrogen bonds with Asp100, His141, and Tyr304. The residues within the 6 Å distance to its binding ligand with low ConSurf grade (≤7) are shown in sticks. The residues for Zn atom binding (Asp177, His179, and Asp265) and for catalyzing histone deacetylase (His141, His142, and Tyr304) are highly conserved. Hence, their functions for histone deacetylase are conserved [15]. Gly28 and Glu99 are in low ConSurf grades in all the HDAC classes. For three major classes, some residues of Class I are lost in Class IV, even the residues near vorinostat, such as Phe206 and Glu99. Class I (4) has fewer residues in low ConSurf grades than Class II (7). For subclasses of Class I, HDAC1 (5) has more residues in low ConSurf grade than HDAC3 (3), but fewer than HDAC8 (8). For subclasses of Class II, some gaps (colored in yellow) are found in each subclass, and the gap residues of HDAC4 and HDAC12 are near the binding ligand. In summary, the binding domain of Class I is most complete and conserved, especially HDAC1 and HDAC3. It may be the reason for their higher drug ability in humans [9]. Meanwhile, the difference in subclasses of HDACs may be used for the highly selective drug or pesticide design.”.
